# REVISITING STRUCTURED DROPOUT

## ABSTRACT

Large neural networks are often overparameterised and prone to overfitting, Dropout is a widely used regularization technique to combat overfitting and improve model generalization. However, unstructured Dropout is not always effective for specific network architectures and this has led to the formation of multiple structured Dropout approaches to improve model performance and, sometimes, reduce the computational resources required for inference. In this work, we revisit structured Dropout comparing different Dropout approaches to natural language processing and computer vision tasks for multiple state-of-the-art networks. Additionally, we devise an approach to structured Dropout we call ***ProbDropBlock*** which drops contiguous blocks from feature maps with a probability given by the normalized feature salience values. We find that with a simple scheduling strategy the proposed approach to structured Dropout consistently improved model performance compared to baselines and other Dropout approaches on a diverse range of tasks and models. In particular, we show ***ProbDropBlock*** improves RoBERTa finetuning on MNLI by $0.22\%$, and training of ResNet50 on ImageNet by $0.28\%$.

## 1 INTRODUCTION

In our modern society, Deep Neural Networks have become increasingly ubiquitous, having achieved significant success in many tasks including visual recognition and natural language processing Heaton (2020); Jumper et al. (2021); Schrittwieser et al. (2020). These networks now play a larger role in our lives and our devices, however, despite their successes they still have notable weaknesses. Deep Neural Networks are often found to be highly overparameterized, and as a result, require excessive memory and significant computational resources. Additionally, due to overparameterization, these networks are prone to overfit their training data.

There are several approaches to mitigate overfitting including reducing model size or complexity, early stopping Caruana et al. (2000), data augmentation (DeVries & Taylor, 2017) and regularisation (Loshchilov & Hutter, 2017). In this paper, we focus on Dropout which is a widely used form of regularisation proposed by Srivastava et al. (2014b). Standard Unstructured Dropout involves randomly deactivating a subset of neurons in the network for each training iteration and training this subnetwork, at inference time the full model could then be treated as an approximation of an ensemble of these subnetworks.

Unstructured Dropout was efficient and effective and this led to it being widely adopted, however, when applied to Convolutional Neural Networks (CNNs), unstructured Dropout struggled to achieve notable improvements He et al. (2016); Huang et al. (2017) and this led to the development of several structured Dropout approaches Ghiasi et al. (2018); Dai et al. (2019); Cai et al. (2019) including DropBlock and DropChannel. DropBlock considers the spatial correlations between nearby entries in a feature map of a CNN and attempts to stop that information flow by deactivating larger contiguous areas/blocks, while DropChannel considers the correlation of information within a particular channel and performs Dropout at the channel level. However, since the development of these structured approaches, there have been further strides in network architecture design, with rising spread and interest in Transformer-based models.

Given the success achieved by block-wise structured Dropout on CNNs, it is only natural to ask the question, *do these approaches apply to Transformer-based models?* Structured Dropout approaches for transformers seem to focus on reducing the model size and inference time, these works place

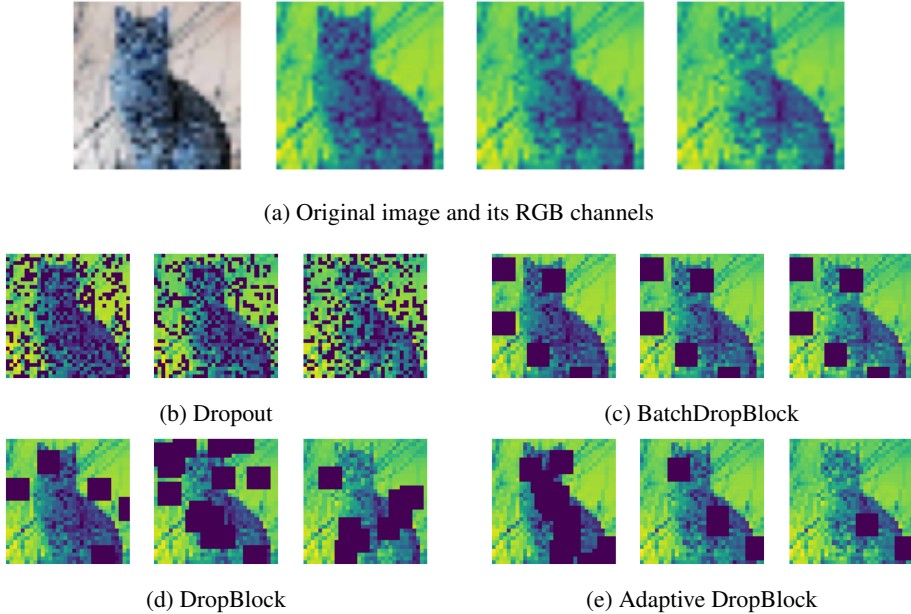

(a) Original image and its RGB channels

(b) Dropout

(c) BatchDropBlock

(d) DropBlock

(e) Adaptive DropBlock

Figure 1: An illustration of applying different Dropouts to an image.

more emphasis on pruning or reducing computational resources Xin et al. (2020); Fan et al. (2019) than combating overfitting which is the focus of this paper.

In this paper, we revisit the idea of structured Dropout for current state-of-the-art models on language and vision tasks. Additionally, we devised our own form of adaptive structured Dropout - ***ProbDropBlock*** and compare it to preexisting approaches to structured and unstructured Dropout.

In Figure 1 we illustrate the effects of select structured and unstructured Dropout approaches on an image of a cat. As can be seen in Figure 1a the original image consists of three channels (RGB) which are aggregated to form the image. Different approaches to Dropout may treat channels differently. In Figure 1b we illustrate the effect of unstructured Dropout on this image, the many small black squares represent deactivated/dropped weights at a pixel level and we also see different pixels have been deactivated in each channel. In Figure 1c we see fewer but larger black squares, and that the locations of dropped pixels are consistent between channels, however, this is not the case in Figure 1e and Figure 1d. In this work, we say that BatchDropBlock is channel consistent i.e. channels do not deactivate blocks independently rather the deactivated blocks are consistent between channels.

In Figure 1b, Figure 1c, Figure 1d, for a single channel there is a uniform probability of any pixel or block (depending on the approach) to be dropped and so deactivated pixels may not contain any of the key information required to identify this image as a cat (i.e. the probability of deactivating a pixel/block belonging to the cat is the same as that of one belonging to the background). This is not the case for Figure 1e, in our adaptive DropBlock approach the probability of a block being dropped is dependent on the value of the center pixel in the block. It can be seen that this approach is not channel consistent and deactivated pixels are concentrated on the cat.

Figure 1 is illustrative to give one an intuitive understanding of these techniques, as in practice these techniques are applied to feature maps which are the output activations of a preceding layer of the network. The contributions of this paper include:

- The testing of preexisting unstructured and structured Dropout approaches on current state-of-the-art models including transformer-based models on natural language inference and vision tasks. We reveal that structured Dropouts are generally better than unstructured ones on both vision and language tasks.

- The proposal of a new approach to structured dropout named ProbDropBlock, which improved model performance on both vision and language tasks. ProbDropBlock is adaptive and the blocks dropped are dependent on the relative per-pixel values. It improves RoBERTa finetuning on MNLI by $0.22\%$ and ResNet50 on ImageNet by $0.28\%$.

- Further observation of the benefits of simple linear scheduling observed Ghiasi et al. (2018) for both structured and unstructured Dropout on a range of vision and language models.

## 2 RELATED WORK

In this section, we briefly review related works in the areas of structured and unstructured Dropouts used as both a regularization technique to improve model performance and as an approach to pruning to reduce the model size and computational requirements. We briefly detail unstructured Dropout and the various structured Dropouts devised for other network architectures.

### 2.1 UNSTRUCTURED DROPOUT

To help address the problem of overfitting in neural networks, Srivastava et al. (2014a) proposed Dropout as a simple way of limiting the co-adaptation of the activation of units in the network. By randomly deactivating units during training they sample from an exponential number of different thinned networks and at test time an ensemble of these thinned networks is approximated by a single full network with smaller weights. Dropout led to improvements in the performance of neural networks on various tasks and has become widely adopted. This form of Dropout in this work we refer to as unstructured Dropout as any combination of units in the network may be randomly dropped/deactivated. In the following subsection, we consider forms of structured Dropout which extend this idea further for other network architectures and tasks.

### 2.2 DROPBLOCK AND OTHER STRUCTURED DROPOUTS

Ghiasi et al. (2018) proposed DropBlock as a way to perform Structured Dropout for Convolutional Neural Nets (CNNs). They suggest that unstructured Dropout is less effective for convolutional layers than fully connected layers because activation units in convolutional layers are spatially correlated so information can still flow through convolutional networks despite Dropout and so they devised DropBlock which drops units in a contiguous area of the feature map collectively. This approach was inspired by Devries & Taylor (2017)'s Cutout, a data augmentation method where parts of the input examples are zeroed out. DropBlock generalized Cutout by applying Cutout at every feature map in convolutional networks. Ghiasi et al. (2018) also found that a scheduling scheme of linearly increasing DropBlock's zero-out ratio performed better than a fixed ratio.

Dai et al. (2019) extended DropBlock to Batch DropBlock. Their network consists of two branches; a global branch and a feature-dropping branch. In their feature dropping branch they randomly zero out the same contiguous area from each feature map in a batch involved in computing loss function. They suggest zeroing out the same block in each batch allows the network to learn a more comprehensive and spatially distributed feature representation.

Larsson et al. (2016) proposed DropPath in their work on FractalNets. Just as Dropout prevents the co-adaptation of activations, DropPath prevents the co-adaptation of parallel paths in networks such as FractalNets by randomly dropping operands of the join layers. DropPath provides at least one such path while sampling a subnetwork with many other paths disabled. DropPath during training alternates between a global sampling strategy which returns only a single path and a local sampling strategy in which a join drops each input with fixed probability, but with a guarantee, at least one survives. This encourages the development of individual columns as performant stand-alone subnetworks.

Cai et al. (2019) proposed DropConv2d as they suggest the failure of standard dropout is due to conflict between the stochasticity of unstructured dropout and the following Batch Normalization (BN) step. They propose placing dropout operations right before the convolutional operation instead of BN or replacing BN with Group Normalization (GN) to reduce this conflict. Additionally, they devised DropConv2d which draws inspiration from DropPath and DropChannel, they treat each

channel connection as a path between input and output channels and perform dropout on replicates of each of these paths.

DropBlock, BatchDropBlock, DropPath and DropConv2d are forms of structured Dropout designed with specific architecture in mind. However, as seen by Cai et al. (2019) DropConv2d an approach to structured Dropout designed for a given network can still be useful to novel network architecture. Aside from being used to improve generalization, structured Dropout has also been used as an approach to pruning and reducing computational resource requirements at inference time.

Fan et al. (2019) proposed LayerDrop as a means of regularization for transformers during training and efficient pruning at inference time reducing the large amount of computation these models require. This approach is a form of structured Dropout where instead of deactivating weights independently throughout the network, weights that collectively form a single structure in the network are deactivated. Attention heads in a transformer are typically computed in parallel, as a result in the paper the structure they focused on deactivating where fully connected layers. Using this approach they found that they were able to select sub-networks of any depth from one large network that without finetuning achieved similar performance.

Other forms of run-time structured pruning include (Xin et al., 2020; Gao et al., 2018; Wu et al., 2018), Xin et al. (2020) proposed DeeBERT which accelerates inferencing in BERT models by allowing samples to exit earlier without passing through the entire model under certain conditions as they believe that, for BERT, features provided by the intermediate transformer layers may suffice to classify some input samples. Gao et al. (2018) preserve the full network structure of CNNs and accelerates convolution by dynamically skipping unimportant input and output channels determined by a saliency criterion exploiting the fact that the importance of features computed by convolutional layers is highly input-dependent. Wu et al. (2018) propose BlockDrop where an RL agent learns which blocks in a ResNet to select dynamically for a given novel input image.

In this work, we are interested in dropping structured patterns in the spatial dimension and would like to understand how this style of structured Dropouts can affect current mainstream models such as Transformers and Vision Transformers. We mainly compare our proposed approach with DropBlock and a revised version of BatchDropBlock, since these two methods are dropping at the same granularity as us.

## 3    METHOD

In this section we detail the approach to structured dropout we employed in this paper. Our approach is a form of adaptive DropBlock we call ProbDropBlock, it is inspired by DropBlock as it also randomly removes larger blocks from each feature map, but rather than assigning a uniform probability to each element of being the center point of the block it assigns higher probability to elements with higher activation values.

As such the block removed depends on the model's learned representation of the feature map and we believe this encourages the model to learn a more balanced and diverse feature map. We also employ a simple linear schedule where we linearly increase the base probability $\alpha$ of dropping a block. Ghiasi et al. (2018) made an observation in their paper that this approach can significantly improve performance and is more robust.

In Algorithm 1 we detail our adaptive DropBlock method, the algorithm takes as input the output activations from a layer $A$, a block size $B$, base drop probability $\alpha$ and the $mode$ of the network.

When not in inference mode the algorithm computes a drop ratio $\gamma_{i,j}$ for each entry in the feature map. This is a normalized drop ratio as illustrated in Line 6 of Algorithm 1: this drop ratio is equal to ratio of the absolute value of the entry $abs(A_{i,j})$ and the average of absolute values of all entries in the feature map $\frac{\|A\|_0}{\|A\|_1}$.

Each drop ratio $\gamma_{i,j}$ is multiplied with the base drop probability and constrained to range $[0, 1]$ to give the drop probabilities $q_{i,j}$ for each entry in the feature map as illustrated in Line 7 of Algorithm 1. As a result, entries with values that are higher than the mean absolute entry value for the feature map have a higher probability of being dropped.

---

**Algorithm 1** Adaptive DropBlock

---

1: **procedure** PROBDROPBLOCK($A, B, \alpha, mode$)
2:     **Inputs** Layer Output Activations - $A$, Block Size - $B$, Base Drop Probability - $\alpha$, $mode$
3:     **if** $mode == Inference$ **then**
4:         **return** $A$
5:     **else**
6:         $\gamma_{i,j} = \dfrac{\|A\|_0 \times abs(A_{i,j}))}{\|A\|_1}$         ▷ Compute drop ratio $\gamma_{i,j}$ for each element in A
7:         $q_{i,j} = \min((\alpha \times \gamma_{i,j}), 1)$         ▷ Compute drop probabilities $q_{i,j}$
8:         $M : M_{i,j} \sim Bernoulli(1 - q_{i,j})$         ▷ Randomly sample mask $M$
9:         **for** $M_{i,j}$ in $M$ **do**
10:             **if** $M_{i,j} == 0$ **then**
11:                 **store**$((i, j))$         ▷ Store mask indices with zero entries
12:         $\beta_{lb} = floor\left(\dfrac{B-1}{2}\right)$         ▷ Compute lower bound buffer zone for mask
13:         $\beta_{ub} = round\left(\dfrac{B-1}{2}\right)$         ▷ Compute upper bound buffer zone for mask
14:         **for** $(i, j)$ in **store do**
15:             $M_{i-\beta_{lb}:i+\beta_{ub}, j-\beta_{lb}:j+\beta_{ub}}. = 0$         ▷ Set values in square centered at $M_{i,j}$ to 0
16:         $A = A \times M$         ▷ Apply mask $M$ to $A$
17:         $A = A \times \dfrac{\mathbf{sum}(M)}{\|M\|_0}$         ▷ Normalize
18:     **return** $A$

---

Using drop probabilities $q_{i,j}$ we sample a mask $M$, we modify the mask by constructing a block of size $B$ around each zero entry in the mask and setting values in the box to zero to create a larger contiguous block of zeros. We handle even block sizes by effectively shifting the block's center point by half an entry right.

After modification this mask $M$ is applied to the feature map $A$ by element wise matrix multiplication. Finally, we re-normalize $A$ by the ratio of non-zero entries in $M$ to the total number of entries in $M$ and return it, this is the same re-normalization technique used in DropBlock (Ghiasi et al., 2018). The linear scheduling is implemented by gradually increasing the base drop probability $\alpha$ over the training cycle. The details and effect of the linear scheduling are evaluated in details in Section 4.2.

## 4   EVALUATION

In the following section, we lay out the experimental setup for this paper and briefly discuss the datasets and models considered in Section 4.1.

We then address the importance of probability scheduling for both structured and unstructured Dropout methods in Section 4.2. In Section 4.3, we demonstrate that structured Dropouts generally outperform their unstructured counterpart for both vision and language tasks.

To properly assess our approach to other structured Dropouts we test on both vision and language tasks for state-of-the-art models in Section 4.4 and Section 4.5. We compare the performance of models trained with the proposed ProbDropBlock approach to those trained with other forms of Dropout and baseline models trained without any forms of Dropout.

### 4.1   EXPERIMENT SETUP

In this work, we considered 6 datasets, 3 NLP (natural language processing) datasets and 3 CV (computer vision) datasets.

The NLP datasets considered are all part of **GLUE** - the multitask benchmark and analysis platform for natural language understanding (Wang et al., 2018), we consider MNLI, QNLI and RTE in our evaluation.

For the CV datasets, we consider CIFAR10, CIFAR100 (cif) and ImageNet (Deng et al., 2009) classification. More details about these datasets can be found in Appendix B.

For finetuning the RoBERTa model, we use the hyperparameter setup in Liu et al. (2019). For training the ResNet family models, we train all models using the Adam optimizer Kingma & Ba (2014) and pick the best learning rate from $\{1e^{-5}, 5e^{-5}, 1e^{-4}\}$. For the pyramid vision transformer (PVT-V2) model, we use the standard setup described in Wang et al. (2022). We slightly changed the augmentations in the original PVT-V2 setup, removing Mixup and Random Erasing, so that it is more closely aligned with the ResNet training setup for a better comparison. In the ResNet family and PVT-V2 models, we insert the Dropout mechanism after each residual block. For RoBERTa, we add Dropout, DropBlock or ProbDropblock to the end of each encoder layer.

We run each data point 3 times with different random seeds, and report both the average and standard deviations, the details of our hardware system setup can be found in Appendix D. We picked the dropping Block Size ($B$) to be $B = 4$, and show an ablation of this parameter in Appendix F.

## 4.2 PROBABILITY SCHEDULING

Table 1: Structured (BatchDropBlock) and non-structured Dropouts on CIFAR-10. with and without the linear scheduling on the dropping probability. BDB is BatchDropBlock. $\Delta$ shows the difference between the accuracy compared to baseline, the baseline accuracy on this task is $94.37\%$.

| Method | Dropout | Dropout-Schedule | BDB | BDB-Schedule |
|---|---|---|---|---|
| Resnet50 Acc ↑ | $94.50 \pm 0.04$ | $94.70 \pm 0.14$ | $93.11 \pm 0.22$ | $94.77 \pm 0.29$ |
| $\Delta$ ↑ | $+0.13$ | $+0.33$ | $-1.26$ | $+\mathbf{0.41}$ |

Ghiasi et al. mentioned in their experiments that DropBlock with a linear dropping scheme that decreases the value of keep probability from 1 to $1 - \alpha$ can significantly improve the performance. We test $\alpha \in \{0.1, 0.2, 0.3, 0.5\}$, and pick the best performing $\alpha$ ($\alpha = 0.2$ in this case). A detailed explanation of how we pick the keep probability is in Appendix E.

In Table 1, we apply this linear dropping scheme to both the standard Dropout (Srivastava et al., 2014b) and BatchDropBlock (Dai et al., 2019). We consider a modified version of BatchDropBlock, where all channels of a single input are dropped consistently, but datapoints in a batch can drop independently. One major observation from Table 1 is that an appropriate probability scheduling improves the performance of both structured and unstructured Dropout methods. The scheduling shows a greater impact on structured Dropout.

Intuitively, Dropout servers as a regularization method, and applying it at the start of the training interferes with the optimization; this type of regularization method should be introduced at a later stage of training when the training accuracy starts to become larger than the validation accuracy, or in other words when overfitting starts to arise.

In general, we observed that:

- Structured Dropout (*eg.* DropBlock) with a linear dropping scheme of decreasing the value of keep probability can significantly improve the performance, this aligns with the observation made by Ghiasi et al..

- ***Non-structured Dropout also benefits from the linear dropping scheme***, although it is a less significant improvement than the structured Dropout.

## 4.3 STRUCTURED AND NON-STRUCTURED DROPOUTS

In this section, we compare the performance of the standard Dropout, various structured Dropout schemes (BatchDropBlock and DropBlock) and ProbDropBlock. All methods have a linear dropping scheme of decreasing the value of the keep probability from 1 to $1 - \alpha$. We experimented also

$\alpha \in \{0.1, 0.2, 0.3, 0.5\}$, and used $\alpha = 0.2$ for ResNet and $0.1$ for RoBERTa models, an ablation study of different dropping probabilities can be found in Appendix E.

Table 2: A comparison of the performance of Dropout and various Structured Dropout schemes. BDB is BatchDropBlock. DropBlock is not channel consistent (blocks deactivated are not consistent between channels), ProbDropBlock additionally has dropping probabilities correlated to pixel-wise saliency. RoBERTa is evaluated on MNLI with a baseline accuracy of $87.60\%$, and ResNet50 is evaluated on CIFAR10 with a baseline accuracy of $94.37\%$. $\Delta$ is the difference between the current accuracy and baseline.

| Method | Dropout | BDB | DropBlock | ProbDropBlock |
|---|---|---|---|---|
| Resnet50 Acc ↑ | $94.70 \pm 0.14$ | $94.77 \pm 0.29$ | $95.05 \pm 0.21$ | $94.73 \pm 0.19$ |
| $\Delta$ ↑ | $+0.33$ | $+0.41$ | $+\mathbf{0.68}$ | $+0.35$ |
| RoBERTa Acc ↑ | $87.51 \pm 0.08$ | $87.39 \pm 0.29$ | $87.71 \pm 0.24$ | $87.83 \pm 0.15$ |
| $\Delta$ ↑ | $-0.09$ | $-0.21$ | $+0.11$ | $+\mathbf{0.22}$ |

Language models such as BERT (Devlin et al., 2018) and RoBERTa (Liu et al., 2019) are based on the multi-head attention mechanism (Vaswani et al., 2017). Prior work has not studied how to apply coarse-grained Dropout techniques on transformer-based architectures.

We apply these structured Dropouts in a head-wise manner, this means for BatchDropBlock, we drop the same pattern across heads in multi-head attention. DropBlock, in contrast, then drops each head independently.

The original RoBERTa used a Dropout with $\alpha = 0.1$, and we replaced all of these Dropouts with the regularization strategies shown in Table 2. Notice, in this case, our baseline considered is a standard RoBERTa without any Dropouts. Our RoBERTa baseline on MNLI achieves $87.60\%$. The baseline accuracy for ResNet50 on CIFAR10 is $94.37\%$.

The ReNet50 model is evaluated on CIFAR10. The striding of the network is adjusted to fit into this smaller image size of CIFAR10. The details of this network architecture are summarized in Appendix C. Table 2 confirmed with the observation made by Ghiasi et al. (2018) that structured Dropouts are generally better than standard, unstructured Dropouts on vision tasks. In addition, our results in Table 2 also suggest that structured Dropout is better on MNLI.

Another interesting observation is that unstructured Dropout does not provide any performance gains for language models ($-0.09$ on RoBERTa). In the meantime, we see that BDB, although works reasonably well on ResNet50, has a detrimental impact on the performance of RoBERTa. This means that applying structured Dropout methods to each Transformer head independently is important for these methods to improve the performance of language models. We will investigate this phenomenon in greater detail in Section 4.4.

In general, we observed that:

- In addition to what was originally shown by Ghiasi et al., we observed ***structured Dropout techniques are generally better not only on vision tasks but also on language tasks.***
- Structured Dropout techniques are more advantageous on language tasks compared to vision tasks.

## 4.4 LANGUAGE TASKS

Table 3 demonstrates the performance of RoBERTa finetuned on three GLUE tasks (MNLI, QNLI, RTE) using different structured Dropout techniques. To our best knowledge, we are the first to investigate the effect of block-wise structured Dropout methods on Transformer-based models.

We observe that both DropBlock and ProbDropBlock consistently outperform BatchDropBlock (BDB) and that ProbDropBlock is the best performing method. Both DropBlock and ProbDropBlock can significantly help RoBERTa achieve better performance, while BDB has a negative impact on its accuracy.

The major difference with BDB is that each head drops blocks with the same pattern across heads, we observe this is having a negative effect on the performance of RoBERTa.

The proposed strategy, ProbDropBlock, is able to achieve the best performance on all tasks and is able to outperform the original RoBERTa model by a significant margin. This is a clear indication that dropping structured patterns based on the saliency values of the attention maps is advantageous.

Table 3: Different Structured Dropout schemes. DropBlock is channel independent (channels are not dropped independently), ProbDropBlock additionally has dropping probabilities correlated to element-wise saliency. The RoBERTa model is first pretrained on a large unlabeled text corpus and subsequently finetuned on these tasks, which is the same setup in Liu et al..

| Method | Metric | MNLI | QNLI | RTE |
|---|---|---|---|---|
| Baseline | Accuracy $\uparrow$ | $87.60 \pm 0.04$ | $92.75 \pm 0.03$ | $73.28 \pm 0.02$ |
| BatchDropBlock | Accuracy $\uparrow$ | $87.39 \pm 0.29$ | $92.70 \pm 0.06$ | $70.64 \pm 0.07$ |
| | $\Delta \uparrow$ | $-0.21$ | $-0.05$ | $-2.64$ |
| DropBlock | Accuracy $\uparrow$ | $87.71 \pm 0.24$ | $92.81 \pm 0.11$ | $72.51 \pm 0.08$ |
| | $\Delta \uparrow$ | $+0.11$ | $+0.06$ | $-0.77$ |
| ProbDropBlock | Accuracy $\uparrow$ | $87.83 \pm 0.15$ | $92.90 \pm 0.12$ | $74.25 \pm 0.03$ |
| | $\Delta \uparrow$ | $\mathbf{+0.22}$ | $\mathbf{+0.15}$ | $\mathbf{+0.97}$ |

In general, we made the following observations from Table 3:

- BatchDropBlock generally has a negative impact on the performance of language models.
- Structured Dropout applied identically per head (BatchDropBlock) does not improve model performance on language tasks, both DropBlock and ProbDropBlock outperform BDB by a significant margin. ***Dropping heads independently is critical for better performance on language models.***
- The multi-head attention modules in the transformer benefit the most from ProbDropBlock.

### 4.5 VISION TASKS

Table 4: Different Structured Dropout schemes. DropBlock is channel independent (channels are not dropped independently), ProbDropBlock additionally has dropping probabilities correlated to element-wise saliency. ResNet50 and WideResNet28 are from the ResNet family with adjusted striding to match the CIFAR image size. PVTv2-B1 is the pyramid vision transformer.

| Method | Metric | CIFAR10 | | CIFAR100 | |
| | | ResNet50 | PVTv2-B1 | WideResNet28 | PVTv2-B1 |
|---|---|---|---|---|---|
| Baseline | Accuracy $\uparrow$ | $94.37 \pm 0.32$ | $95.59 \pm 1.00$ | $74.72 \pm 0.08$ | $82.38 \pm 0.19$ |
| BatchDropBlock | Accuracy $\uparrow$ | $94.77 \pm 0.29$ | $95.99 \pm 0.15$ | $74.97 \pm 0.26$ | $82.22 \pm 0.34$ |
| | $\Delta \uparrow$ | $+0.41$ | $+0.40$ | $+0.25$ | $-0.16$ |
| DropBlock | Accuracy $\uparrow$ | $95.05 \pm 0.21$ | $95.89 \pm 0.09$ | $74.99 \pm 0.08$ | $82.26 \pm 0.35$ |
| | $\Delta \uparrow$ | $\mathbf{+0.68}$ | $+0.30$ | $+0.27$ | $-0.12$ |
| ProbDropBlock | Accuracy $\uparrow$ | $94.73 \pm 0.19$ | $96.15 \pm 0.01$ | $75.13 \pm 0.27$ | $82.44 \pm 0.16$ |
| | $\Delta \uparrow$ | $+0.35$ | $\mathbf{+0.56}$ | $\mathbf{+0.41}$ | $\mathbf{+0.06}$ |

Table 5: ProbDropBlock and baseline performance for ImageNet classification.

| Method | Metric | ImageNet | |
| | | ResNet50 | PVTv2-B1 |
|---|---|---|---|
| Baseline | Accuracy $\uparrow$ | $74.22 \pm 0.06$ | $78.27 \pm 0.04$ |
| ProbDropBlock | Accuracy $\uparrow$ | $74.50 \pm 0.17$ | $78.88 \pm 0.22$ |
| | $\Delta \uparrow$ | $+0.28$ | $+0.61$ |

Table 4 and Table 5 demonstrate the results of applying different structured Dropout methods on CIFAR10, CIFAR100 and ImageNet. Vision Transformers (ViTs) recently have demonstrated great capabilities on major vision benchmarks (Liu et al., 2021; Wang et al., 2022), so we consider both the ResNet family (He et al., 2016; Zagoruyko & Komodakis, 2016) and Pyramid Vision Transformer

Wang et al. (2022) in our experiment. Prior research has hardly systematically studied the effect of structured Dropouts on Vision Transformer models.

We observe that in general ProbDropBlock shows the best performance on all dataset network combinations, except for one outlier which is ResNet50 on CIFAR10. Interestingly, we observe a phenomenon that BDB (BatchDropBlock) in general improves the performance of both CNNs and ViTs according to Table 4. This is very different from the phenomenon observed in Section 4.4 that BDB generally decreases the performance of language models.

We also notice that only ProbDropBlock can slightly increase the accuracy of PVTv2-B1 on CIFAR100. We then realised that PVTv2-B1 has a validation accuracy that is not greatly larger than its training accuracy, meaning that this model does not overfit the dataset by a significant margin. For instance, ResNet50 on CIFAR10 overfits heavily on the CIFAR10 task and thus it benefits the most from regularization methods. The PVTv2-B1 model architecture used for both CIFAR10 and CIFAR100 is significantly smaller than the original model, the details of this model architecture difference are explained in Appendix C. In addition, CIFAR100 is a harder task than CIFAR10, we see PVTv2-B1 benefits less from regularization methods such as structured Dropouts when the model is not overfitting.

Table 4 generally demonstrate that ProbDropBlock is the best regularization method on three out of the four model-dataset combinations. When we train PVTv2-B1 on CIFAR100, both BDB and DropBlock fail to improve the model performance since the model does not overfit greatly to the dataset; but, ProbDropBlock still makes a positive impact on model performance. We further tested the effect of ProbDropBlock on ImageNet and Table 5 summarizes our results. We demonstrate that ProbDropBlock is an effective regularization method, it provides us $+0.28\%$ and $+0.61\%$ accuracy gains for ResNet50 and PVTv2-B1 respectively.

Structured Dropout techniques in general can help vision models based on our observations on Table 4 and Table 5. We make the following observations:

- BatchDropBlock has a positive impact on computer vision models, this is different from what we have observed in Section 4.4.
- Models that are not overfitting benefit less from structured Dropouts.
- Structured Dropout methods generally help vision models to learn better, ***the proposed ProbDropBlock is effective on both CNNs and ViTs.***

## 5 CONCLUSION

In this paper, we revisit the ideas of structured Dropout for current state-of-the-art models, we devise our form of adaptive structured Dropout - ProbDropBlock and compare preexisting structured and unstructured Dropout approaches to ours on vision and language tasks. We demonstrated the utility of a simple linear dropping schedule for both structured and unstructured Dropouts supporting a similar observation made by Ghiasi et al. (2018).

Our approach, ProbDropBlock was able to achieve improvements in performance for all networks and task combinations and outperformed other forms of Dropout considered for the majority of combinations we evaluated. In particular, ProbDropBlock improved RoBERTa finetuning on MNLI by $0.22\%$, and training of ResNet50 on ImageNet by $0.28\%$.

This work demonstrates the utility of structured Dropout approaches not just on residual networks and CNNs, but on language and vision transformers. However, there is a limit to the gains achievable through Dropout alone, as demonstrated by the results of the PVTv2-B1 model on the CIFAR-100 dataset, when there is minimal overfitting in the model regularization only provides minimal gain.

## 6 ETHICS STATEMENT

Language and vision models typically require a significant amount of data for training and testing. Our work only uses datasets that are widely used within the ML community, however, there are some ethical concerns about the collection methodologies and entries in some of these datasets Asano et al. (2021). We make use of these datasets because they are established and recognized benchmarks while we acknowledge possible ethical issues with these datasets. In this work, we focus on improving generalization by combating overfitting through adaptive structured Dropout. Generally, improved model generalization is a positive outcome but that assumes these trained models are not applied to nefarious purposes, however, we can not ensure this.

## 7 REPRODUCIBILITY STATEMENT

We discuss how we setup our experiments in Section 4.1. We explained the learning rate and optimizer setup in that section. At the beginning of Section 4.2, Section 4.4 and Section 4.5, we explained the choice of our probability scheduling and the picked $\alpha$ value. In Appendix E and Appendix F, we explained how we picked the hyperparameters $\alpha$ and $B$ based on these ablation studies. Each of our experiments is repeated for 3 times, and both the arithmetic mean and standard deviation are reported.

Appendix B details the setup of each dataset and Appendix C details the models we have used and included any modifications we have made to these models. Our hardware system used to perform experiments is reported in Appendix D.

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

# A MOTIVATION FROM A THEORETICAL LENS

## A.1 MARGINALIZING PROBDROPBLOCK

In considering the effect of our adaptive structured approach to Dropout we present a modified scenario of marginalising out Dropout in a simple linear regression case with a block size of 1. This setup was used in the original Dropout paper Srivastava et al. (2014b) to illustrate how Dropout acts as a regulariser when marginalised out.

In our scenario we take as input the output activations $A \in \mathbb{R}^{N \times M}$ of a layer of a network, a weight vector $\mathbf{w} \in \mathbb{R}^M$ and a target vector of $\mathbf{y} \in \mathbb{R}^N$.

As we all know the aim of linear regression is to minimise $\|\mathbf{y} - A\mathbf{w}\|$. ProbDropBlock randomly drops entries in the matrix $A$ with entries being retained with a probability $p_{i,j} = 1 - (q \times \alpha_{i,j})$ where $\alpha_{i,j}$ gives the adaptive ratio for dropping feature entries based of relative magnitude. $p_{i,j}$ may be rewritten as $p \times h_{i,j}$ where $p$ is a constant keep probability and $h_{i,j}$ is a scalar that modifies p to give the keep probabilities for matrix entry $i, j$.

After ProbDropBlock the input $A$ can be expressed as $R * A$ where $R \in \{0, 1\}^{N \times M}$ is a random matrix and $R_{i,j} \sim \text{Bernoulli}(p_{i,j})$ and $*$ represents element wise matrix multiplication. So marginalising the effect of dropout out of the objective function becomes

$$\mathbb{E}_{R_{i,j} \sim Bernoulli(p_{i,j})} \left[ \|\mathbf{y} - (R * A)\mathbf{w}\|^2 \right] \tag{1}$$

This becomes

$$\mathbf{y}^T\mathbf{y} - \mathbf{w}^T\mathbb{E}\left[(R * A)^T\right]\mathbf{y} - \mathbf{y}^T\mathbb{E}\left[(R * A)\right]\mathbf{w} + \mathbf{w}^T\mathbb{E}\left[(R * A)^T(R * A)\right]\mathbf{w} \tag{2}$$

It is clear that the $\mathbb{E}\left[(R * A)^T\right] = \mathbb{E}\left[(R * A)\right]^T$ and $\mathbb{E}\left[(R * A)\right] = P * A$ where $P$ is a matrix of the keep probabilities for each entry $(i, j)$. So $\mathbb{E}\left[(R * A)\right]$ may be rewritten as $p\tilde{A}$ where $\tilde{A} = H * A$, $p$ is a constant and $H$ is a re-scaling matrix of entries $h_{i,j}$ that modify the constant $p$ as needed.

$\mathbb{E}\left[(R * A)^T(R * A)\right]$ is a little more involved but works out to be $p^2\tilde{A}^T\tilde{A} + (p - p^2)\text{diag}\left(\tilde{A}^T\tilde{A}\right)$.

And so the full expectation works out to be

$$\mathbf{y}^T\mathbf{y} - p\mathbf{w}^T\tilde{A}^T\mathbf{y} - p\mathbf{y}^T\tilde{A}\mathbf{w} + p^2\mathbf{w}^T\tilde{A}^T\tilde{A}\mathbf{w} + (p - p^2)\mathbf{w}^T\text{diag}\left(\tilde{A}^T\tilde{A}\right)\mathbf{w} \tag{3}$$

Which can be rewritten as

$$\|\mathbf{y} - p\tilde{A}\mathbf{w}\|^2 + (p - p^2)\|\text{diag}\left(\tilde{A}^T\tilde{A}\right)^{\frac{1}{2}}\mathbf{w}\|^2 \tag{4}$$

The second term is never negative as $p \geq p^2$ because $p \leq 1$ and so the second term acts as a regulariser when we try to minimise this expression with respect to $\mathbf{w}$ as is the case in linear regression. In fact this equation takes the same form as that seen in the original Dropout paper Srivastava et al. (2014b), with the exception of an $\tilde{A}$ instead on an $A$.

Effectively the modifier matrix $H$ is re-scaling the input features scaling down entries with higher than average values and scaling up entries with lower than average values. While in a neural network dynamics are more complex and the activation outputs are evolving as training progresses we believe this adaptive structured dropout approach encourages the network to learn a more robust feature representation.

## A.2 APPROXIMATING ENSEMBLES USING DROPOUT

The initial Dropout paper Srivastava et al. (2014b) also states we can consider unstructured Dropout as a way of training exponentially many sparse networks with extensive weight sharing and then at test time we approximately average this ensemble of networks by rescaling the weights.

ProbDropBlock is an adaptive and structured approach to Dropout. In ProbDropBlock instead of uniformly sampling sparse networks from the original network during training we sample sparse networks with lower activations more often. If we assume activations are a proxy for containing relevant information the sampled sparse networks are faced with harder problems are they are missing likely relevant information and so are likely to perform *'worse'* at the same stage of training. So by sampling these *'worse'* performing sparse networks more often during training we provide additional training for these *'worse'* sparse networks and improve the overall performance of the approximated average ensemble at test time.

## B  DATASETS

### B.1  LANGUAGE TASKS

MNLI (Multi-Genre Natural Language Inference) corpus (Williams et al., 2018; Wang et al., 2018) is a collection of sentence pairs with textual entailment annotations gathered via crowd sourcing. The sentences are paired as premise and hypothesis and the task is to predict if the premise entails the hypothesis (*entailment*), contradicts the hypothesis (*contradiction*) or neither (*neutral*). The corpus is modeled on the SNLI (Stanford Natural Lanfuage Inference) corpus (Bowman et al., 2015), but differs in that covers a range of genres of spoken and written text, and supports a distinctive cross-genre generalization evaluation (Williams et al., 2018; Wang et al., 2018). The premises are gathered from ten different sources, including fiction, government reports and transcribed speeches. It consists of 393k train samples and 20k test samples.

QNLI (Question-answering Natural Language Inference) corpus is a dataset automatically derived from SQuAD (Stanford Question Answering Dataset) (Rajpurkar et al., 2016; Wang et al., 2018). SQuAD is a question-answering dataset which consists of question-paragraph pairs, where a sentence in the paragraph contains the answer to the corresponding question. QNLI is constructed by converting the task into sentence pair classification by forming a pair between each question and each sentence in the corresponding context. It consists of 105k training samples and 5.4k testing samples.

RTE RTE The Recognizing Textual Entailment (RTE) datasets is a combination of several datasets which came from a series of annual textual entailment challenges Wang et al. (2018); Dagan et al. (2005); Haim et al. (2006); Giampiccolo et al. (2007); Bentivogli et al. (2009). It consists of 2.5k training samples and 3k testing samples.

### B.2  VISION TASKS

CIFAR-10 & CIFAR-100 The CIFAR-10 dataset contains 60000 32x32 colour images divided into 10 classes, each with 6000 images. There are 50,000 training and 10,000 test images.

The CIFAR-100 dataset is just like the CIFAR-10, except it has 100 classes containing 600 images each. There are 500 training images and 100 testing images per class. The 100 classes in the CIFAR-100 are grouped into 20 superclasses. Each image comes with a "fine" label (the class to which it belongs) and a "coarse" label (the superclass to which it belongs).

The ILSVRC 2012 image classification dataset contains 1.2 million images for training and 50,000 for validation from 1000 classes. The input image sizes are $224 \times 224$ center crop to images at test time. The results are reported on the validation set.

## C  NETWORKS

### C.1  ROBERTA

The RoBERTa model is kept the same as its original form proposed by Liu et al. (2019). In our experiment, we consider the RoBERTa-base model only. The base model contains 12 layers, with a hidden size of 768, an FFN inner hidden size of 3072 and 12 attention heads. The original model uses Dropout and we replace all of the original Dropouts to structured Dropout methods.

### C.1.1 RESNET

We consider both the original ResNet (He et al., 2016) and its wider alternative (WideResNet) (Zagoruyko & Komodakis, 2016). These networks normally have one convolutional layer (named stem) and four other residual blocks. For the CIFAR10 and CIFAR100 classification, we change the striding of the first convolution to 1 and deleted the first max pooling. These adaptions help the network to operate with the $32 \times 32$ image size on CIFAR datasets

### C.1.2 PVT-V2

Table 6 demonstrates the detailed setup of the PVT-V2 structure used for CIFAR and ImageNet tasks. The rest of the setup parameters are the same as Wang et al. (2022), and the setup is the same as the PVTV2-B1 model.

Table 6: PVT-V2 setup for vision datasets, $e$ is the embedding dimension and $s$ is the striding used for the overlapping patch embedding.

| Layer name | CIFAR10/CIFAR1000 | ImageNet |
|---|---|---|
| Stage 1 | $e = 16, s = 4$ | $e = 64, s = 4$ |
| Stage 2 | $e = 32, s = 2$ | $e = 128, s = 2$ |
| Stage 3 | $e = 64, s = 1$ | $e = 256, s = 2$ |
| Stage 2 | $e = 128, s = 2$ | $e = 512, s = 2$ |

### C.1.3 ADDITIONAL RESULTS ON MACHINE TRANSLATION

To illustrate that the proposed dropout scheme also works on Sequence to Sequence tasks, we performed an evaluation of it on IWSLT'14 English (EN) to German (DE) and German (DE) to English (EN).

Table 7: Machine Translation Task (IWSLT'14) using a RoBERTa model, results are reported as BLEU scores.

| | EN $\rightarrow$ DE | DE $\rightarrow$ EN |
|---|---|---|
| Baseline | 27.24 | 33.21 |
| ProbDropBlock | 27.84 (+0.60) | 33.44 (+0.23) |

## D HARDWARE SYSTEM

We used a variety of hardware systems, our initial testing and CIFAR10 results are generated on a hardware system with 4 x NVIDIA GeForce RTX 2080 Ti GPUs. The ImageNet training and RoBERTa training are performed on 4 x Nvidia A100 SXM4 80GB GPUs. The total amount of GPU training cost for all the expriments in this paper is around 20 GPU-days.

## E PICKING THE DROPPING PROBABILITY

Table 8: Different Dropping probabilities for ProbDropBlock.

| Probability | ResNet50 on CIFAR10 | PVT-V2 on CIFAR100 | RoBERTa on MNLI |
|---|---|---|---|
| 0.0 | $94.37 \pm 0.32$ | $82.38 \pm 0.19$ | $87.60 \pm 0.04$ |
| 0.1 | $94.70 \pm 0.14$ | $\mathbf{82.44 \pm 0.16}$ | $\mathbf{87.83 \pm 0.15}$ |
| 0.2 | $\mathbf{94.73 \pm 0.19}$ | $82.21 \pm 0.13$ | $87.32 \pm 0.11$ |
| 0.3 | $94.20 \pm 0.30$ | $82.10 \pm 0.16$ | $69.45 \pm 0.16$ |

## F    PICKING THE BLOCK SIZE

The block size for DropBlock is a hyperparameter that needs to be tuned.    We test $B \in \{2, 4, 6, 8, 10\}$, and pick the best performing $B$ ($B = 4$ in this case).

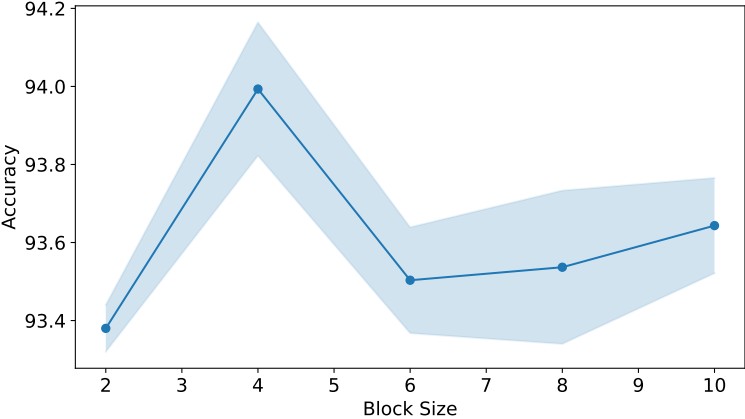

Figure 2: The effect of block size on the performance of DropBlock.

