# OpenReview forum: "Revisiting Structured Dropout"
_ICLR.cc/2023/Conference — Submitted to ICLR 2023_

### Official Review · Reviewer_zLHu · 2022-10-24

**Confidence:** 4
**Clarity, Quality, Novelty And Reproducibility:** 1. the writing is generally clear and…
**Correctness:** 2
**Technical Novelty And Significance:** 2
**Empirical Novelty And Significance:** 2
**Recommendation:** 5

**Details Of Ethics Concerns:**

No ethics concerns.

**Strength And Weaknesses:**

Strengths:

1. The paper proposed an adaptive structured dropout method that is beneficial for both vision and language tasks. It indicates the potential of using structured dropout in Transformer-based models.
2. The paper provided an thorough investigation of existing structured dropout methods.

Weakness:

1. The experimental settings and the performance gains are insufficient to demonstrate the effectiveness of the method. In Table 2, I am not quite convinced by the RoBERTa results, as unstructured dropout can usually achieve some gains for Transformers/ViTs. The authors may present results on more models/tasks to show that structured dropout works better in Transformers/ViTs. For a fair comparison, the authors may compare the proposed adaptive structured dropout to adaptive unstructured dropout. In Table 3, the gains are marginal (typo: the gain on QNLI should be $0.15$).

2. The motivation of assigning higher drop rate to activations with higher magnitude is unclear. More ablations and analysis are needed to understand its influence to the training. It seems to impose a strong regularization but may also prevent the model from seeing important features.

3. Expanding blocks based on the sampled $A_{i,j}$s may be problematic as the surrounding activations may not have high magnitude on average.


**Summary Of The Paper:**

This paper proposed an adaptive structured dropout method, ProbDropBlock, which drops contiguous blocks from feature maps with a probability given by the normalized feature salience values. The authors evaluate ProbDropBlock on both vision and language tasks.

**Summary Of The Review:**

The paper proposed an adaptive structured dropout method that is beneficial for both vision and language tasks. It indicates the potential of using structured dropout in Transformer-based models. However, the experiment results are not sufficient to demonstrate the effectiveness of structured dropout and the proposed adaptive idea, especially in Transformers.

---

> ### Author Response · Authors · 2022-11-15
> **Reply to Reviewer zLHu**
>
> We thank reviewer zLHu for their comments and feedback, we will correct the typos (QNLI) and answer some of the raised concerns.
>
>
> > The motivation of assigning higher drop rate to activations with higher magnitude is unclear. More ablations and analysis are needed to understand its influence to the training. It seems to impose a strong regularization but may also prevent the model from seeing important features.
> >
>
> We actually find this suggestion from the reviewer of forming it as a strong regularization thought provoking and then provided an explanation and attempted a theoretical analysis to ProbDropBlock in our global response ‘Motivation, Explanation and Theoretical Analysis’. We hope this would address the reviewer's concerns.
>
> We have indeed experimented what would happen if the drop ratio is positively correlated, in that case, networks with Dropout actually perform worse than the baseline.
>
> |                             | ResNet50 (CIFAR10) | PVTv2-B1 (CIFAR10) |
> |-----------------------------|--------------------|--------------------|
> | Baseline                    | 94.37              | 95.59              |
> | Positively correlated ratio | 91.33 (-3.04)      | 94.55 (-1.04)      |
>
> > Expanding blocks based on the sampled $A_{i,j}$s may be problematic as the surrounding activations may not have high magnitude on average.
> >
> It is possible that surrounding activations in the block may not be high but block dropout has shown to be useful for image tasks giving the spatial relationship between features. Additionally, we consider block size as an adjustable feature and in the Appendix F we considered the impact of varying block size.
>
> > However, the experiment results are not sufficient to demonstrate the effectiveness of structured dropout and the proposed adaptive idea, especially in Transformers.
> >
>
> We have also added ProbDropBlock to roBERTa now for a Sequence to Sequence task. The task is a translation task (IWSLT14) and we also observed a performance gain.
>
> |               | EN -> DE      | DE -> EN      |
> |---------------|---------------|---------------|
> | Baseline      | 27.24         | 33.21         |
> | ProbDropBlock | 27.84 (+0.60) | 33.44 (+0.23) |

---

### Official Review · Reviewer_Fidd · 2022-10-25

**Confidence:** 3
**Correctness:** 4
**Technical Novelty And Significance:** 3
**Empirical Novelty And Significance:** 2
**Recommendation:** 6

**Clarity, Quality, Novelty And Reproducibility:**

Clarity: very clear
Novelty: sufficiently novel
Reproducibility: fairly straightforward, should be reproducible

**Strength And Weaknesses:**

Pros:
* Exploration of structured dropout for transformers
* An enhancement to drop block that shows better empirical performance as compared to the baseline approach
* Combination with linear dropping schedule gives further gains, reconfirming the value of linear dropout schedule

Cons:
N/A

**Summary Of The Paper:**

This paper proposes application of structured dropout methods (e.g. drop block) to transformers.  It also proposes an enhancement to drop block which utilizes activation values to assign dropout probability to nodes in feature maps as opposed to random selection of blocks.  Empirical results on language and vision tasks demonstrate that structured dropout is advantageous for transformer models and the proposed modification to drop block methods leads to an improvement in accuracy for language and vision tasks.


**Summary Of The Review:**

see comments above

---

> ### Author Response · Authors · 2022-11-15
> **Reply to Reviewer Fidd**
>
> We thank reviewer Fidd for their positive comments and feedback. Please also consider reviewing our general comment about the motivation and analysis for ProbDropBlock.

---

### Official Review · Reviewer_EuKt · 2022-10-27

**Confidence:** 3
**Correctness:** 3
**Technical Novelty And Significance:** 2
**Empirical Novelty And Significance:** 2
**Recommendation:** 5

**Clarity, Quality, Novelty And Reproducibility:**

This paper is easy to follow. The proposed method about adaptive dropout ratio needs further analysis (theoretically or experimentally). Experiment settings can be found in the paper.

**Strength And Weaknesses:**

Strength:

1. The paper is easy to follow

2. The idea of investigating both unstructured and structured dropout methods on SOTA models w.r.t. both NLP and CV tasks is well motivated and beneficial for advance of these two research tasks.

Weakness:

1. Experiment results of baseline method are not SOTA. For example, the (top-1) accuracy of Resnet50 on Imagenet reported in Table 5 is lower than that in Ghiasi et al. (74.22 vs 76.51). Other examples include the accuracy of WideResNet28 on CIFAR100 (which is lower than that reported in the original WRN paper). This weakens the claim that made in the introduction section about testing dropout methods on current SOTA models. Some SOTA results can be found in https://paperswithcode.com/sota/image-classification-on-cifar-100

2. Lack of explanation and analysis about the adaptive dropout ratio, which is negatively correlated to the absolute value of layer output activations. It is hard to understand why the proposed method performs better than existing methods.


**Summary Of The Paper:**

This paper proposes ProbDropBlock, an adaptive version of DropBlock in which the drop probability of the mask is defined to be negatively correlated to the absolute value of the centroid entry. Experiments are conducted on 3 NLP datasets (MNLI, QNLI, RTE) and 3 CV datasets (Cifar-10, Cifar-100, Imagenet).

**Summary Of The Review:**

In general, this paper is well motivated and focuses on an important problem which will be beneficial for future studies in both NLP and CV communities. However, both theoretical and experimental analysis are not thorough and do not support the claims well.

---

> ### Author Response · Authors · 2022-11-15
> **Reply to Reviewer EuKt**
>
> > accuracy of Resnet50 on Imagenet reported in Table 5 is lower than that in Ghiasi et al. (74.22 vs 76.51) … accuracy of WideResNet28 on CIFAR100 (which is lower than that reported in the original WRN paper)
> >
>
> On the ResNet50 side, we would like to clarify that the implementation we used is the post-activation residual ResNet and standard data preprocessing following the original paper [1]:
>
> - There are other later variants of ResNet such as the pre-activation residual ResNet [2], the difference of them is explained in [3] as ResNet and ResNetV2.
> - Although Ghiasi et al. used also the post-activation residual ResNet, we would like to point out that the implementation [4] Ghiasi et al. have used is a modified version.
>     - AutoAug [5] and RandAug [6] have used for a better performance,
>     - The weight initialization and batch normalization has been adjusted.
>
> For the WideResNet case, it is a similar story. The original WResNet paper used ZCA-preprocessing and other forms of preprocessing, that are different from our preprocessing.
>
> We were trying to isolate the benefits you have from data augmentation and network architecture tweaks since it is from a fairly orthogonal optimization dimension. However, we did then implement and ran two models to demonstrate ProbDropBlock can serve as an off-the-shelf method with modified architecture and more complex data augmentation:
>
> |  | RandAug [6] + Pre-act ResNet [2] on ImageNet | TrivialAug [7] + ResNet50 on CIFAR10 |
> | -----------  | ----------- | ----------- |
> | Baseline      | 77.13       | 96.66 |
> | ProbDropBlock   | 77.42 (+0.29)        | 97.02 (+0.36) |
>
> > explanation and analysis about the adaptive dropout ratio
> >
>
> We provided an explanation and attempted a theoretical analysis to ProbDropBlock in our global response ‘Motivation, Explanation and Theoretical Analysis’. We have indeed experimented what would happen if the drop ratio is positively correlated, in that case, networks with Dropout actually perform worse than the baseline.
>
> |                             | ResNet50 (CIFAR10) | PVTv2-B1 (CIFAR10) |
> |-----------------------------|--------------------|--------------------|
> | Baseline                    | 94.37              | 95.59              |
> | Positively correlated ratio | 91.33 (-3.04)      | 94.55 (-1.04)      |
>
>
> [1] Deep Residual Learning for Image Recognition
>
> [2] Identity Mappings in Deep Residual Networks
>
> [3] [https://keras.io/api/applications/resnet/#resnet50v2-function](https://keras.io/api/applications/resnet/#resnet50v2-function)
>
> [4] [https://github.com/tensorflow/tpu/tree/master/models/official/resnet](https://github.com/tensorflow/tpu/tree/master/models/official/resnet)
>
> [5] AutoAugment: Learning Augmentation Policies from Data
>
> [6] RandAugment: Practical automated data augmentation with a reduced search space
>
> [7] TrivialAugment: Tuning-free Yet State-of-the-Art Data Augmentation

---

### Official Review · Reviewer_osqf · 2022-11-04

**Confidence:** 3
**Clarity, Quality, Novelty And Reproducibility:** very  clear, reproducible, good writi…
**Correctness:** 2
**Technical Novelty And Significance:** 2
**Empirical Novelty And Significance:** 2
**Recommendation:** 6

**Strength And Weaknesses:**

strength: clear motivation ,simple method, wide impact

weakness: this manuscript serve well as a systematic review article , but lack of novel in method as a original research article by ICLR standard. The motivation or theoretical analysis of why choosing to conduction stochastic dropout in PROBDROPBLOCK is unclear.

**Summary Of The Paper:**

Structured dropout are methods that determine probability of deactivation based on certain inductive priors such as neighborhood or location, which are important in architectures like CNN.
The author surveyed existing structure dropout method and developed improved version based on block dropout called PROBDROPBLOCK. which assign different deactivation probability based on activation levels. It is interesting to see these method work well in transformers.

**Summary Of The Review:**

Although lacking novelty in method, due to the importance and popularity of dropout in machine learning and the well designed experiments, I think this could bring values to the community

---

> ### Author Response · Authors · 2022-11-15
> **Response to Reviewer osqf**
>
> We thank the reviewer for the comments, and would like to address some of the concerns raised.
>
> > lack of novel in method as a original research article by ICLR standard
> >
>
> We would suggest that we are making several interesting observations that *will change how practitioners are using these dropouts*:
>
> 1. Practitioners today tend to use a single dropout ratio for training. Ghiasi et al. only pointed out that  dropout ratio scheduling is important for structured Dropout, we generalize this claim in this paper and show that  ***Non-structured Dropout (the original Dropout) benefits from the linear dropping scheme.***
> 2. We are the first to systematically demonstrate that S*tructured Dropout techniques are generally better not only on vision tasks but also on language tasks*
> 3. Dropping attention heads independently is critical for better performance on
> language models. ***Batched Dropouts would in general fail in Transformer models.***
>
> We think the practical implications of these observations are meaningful to practitioners in this field.
>
> > The motivation or theoretical analysis
>
> We would like to provide two motivations to our work.
>
> ### Approximating an Ensemble Using Dropout
>
> From the initial Dropout paper we can consider unstructured Dropout as a way of training exponentially many sparse networks with extensive weight sharing and then at test time we approximately average this ensemble of networks.
>
> ProbDropBlock is an adaptive and structured approach to Dropout. In ProbDropBlock instead of uniformly sampling sparse networks from the original network during training we sample sparse networks with lower activations more often.
>
> If we assume activation saliency is correlated with the amount of information that is contained, the sampled sparse networks are now faced with harder problems since they are missing likely relevant information and so likely perform worse. And so by sampling these ‘worse’ performing sparse networks more often during training we provide additional training for these ‘worse’ sparse networks and improve the overall performance of the approximated average ensemble at test time.
>
> Intuitively, we believe ProbDropBlock encourages the sampled sparse network to learn a more expressive feature map and not be overly dependents on particularly high output activations (features).

---

> > ### Author Response · Authors · 2022-11-15
> > **Further part on Motivation**
> >
> > In considering the effect of our adaptive structured approach to Dropout we present a modified scenario of marginalising out Dropout in a simple linear regression case with a block size of 1. This setup was used in the original Dropout paper [1] to illustrate how Dropout acts as a regulariser when marginalised out.
> >
> > In our scenario we take as input the output activations $A \in \mathbb{R}^{N \times M}$ of a layer of a network, a weight vector $\mathbf{w} \in \mathbb{R}^{M}$ and a target vector of $\mathbf{y} \in \mathbb{R}^{N}$.
> >
> > As we all know the aim of linear regression is to minimise  $||\mathbf{y} - A\mathbf{w}||$. ProbDropBlock randomly drops entries in the matrix $A$ with entries being retained with a probability $p_{i,j} = 1 - (q \times \alpha_{i,j})$ where $\alpha_{i,j}$ gives the adaptive ratio for dropping feature entries based of relative magnitude. $p_{i,j}$ may be rewritten as $p \times h_{i,j}$ where $p$ is a constant keep probability and $h_{i,j}$ is a scalar that modifies p to give the keep probabilities for matrix entry $i, j$.
> >
> > After ProbDropBlock the input $A$ can be expressed as $R \ast A$ where $R \in \{0, 1\}^{N \times M}$ is a random matrix and $R_{i,j}$ $\sim$ Bernoulli$(p_{i,j})$ and $\ast$ represents element wise matrix multiplication. So marginalising the effect of dropout out of the objective function becomes
> >
> > \begin{equation}
> >     \mathbb{E}_{R}
> >     \left[
> >     ||\mathbf{y} - \left( R \ast A\right) \mathbf{w}||^{2}
> >     \right]
> > \end{equation}
> >
> > This becomes
> > \begin{equation}
> >     \mathbf{y}^{T}\mathbf{y} -
> >     \mathbf{w}^{T}\mathbb{E}\left[\left( R \ast A\right)^{T}\right]\mathbf{y} -
> >     \mathbf{y}^{T}\mathbb{E}\left[\left( R \ast A\right)\right]\mathbf{w} +
> >     \mathbf{w}^{T}\mathbb{E}\left[\left( R \ast A\right)^{T}\left( R \ast A\right)\right]\mathbf{w}
> > \end{equation}
> >
> > It is clear that the $\mathbb{E}\left[\left( R \ast A\right)^{T}\right] = \mathbb{E}\left[\left( R \ast A\right)\right]^{T}$ and $\mathbb{E}\left[\left( R \ast A\right)\right] = P\ast A$ where $P$ is a matrix of the keep probabilities for each entry $(i,j)$. So $\mathbb{E}\left[\left( R \ast A\right)\right]$ may be rewritten as $p \tilde{A}$ where $\tilde{A} = H \ast A$, $p$ is a constant and $H$ is a re-scaling matrix of entries $h_{i,j}$ that modify the constant $p$ as needed.
> >
> > $\mathbb{E}\left[\left( R \ast A\right)^{T}\left( R \ast A\right)\right]$ is a little more involved but works out to be $p^{2}\tilde{A}^{T}\tilde{A} + (p - p^{2}) \text{diag}\left( \tilde{A}^{T}\tilde{A}\right)$.
> >
> > And so the full expectation works out to be
> > \begin{equation}
> >     \mathbf{y}^{T}\mathbf{y} -
> >     p \mathbf{w}^{T} \tilde{A}^{T}\mathbf{y} -
> >     p \mathbf{y}^{T} \tilde{A}\mathbf{w} +
> >     p^{2} \mathbf{w}^{T}\tilde{A}^{T}\tilde{A}\mathbf{w} +
> >     (p - p^{2}) \mathbf{w}^{T}\text{diag}\left( \tilde{A}^{T}\tilde{A}\right)\mathbf{w}
> > \end{equation}
> >
> > Which can be rewritten as
> > \begin{equation}
> >     ||\mathbf{y} - p\tilde{A}\mathbf{w}||^{2} +
> >     (p - p^{2})||\text{diag}\left( \tilde{A}^{T}\tilde{A}\right)^{\frac{1}{2}} \mathbf{w}||^{2}
> > \end{equation}
> >
> > The second term is never negative as $p \ge p^{2}$ because $p \le 1$ and so the second term acts as a regulariser when we try to minimise this expression with respect to $\mathbf{w}$ as is the case in linear regression. In fact this equation takes the same form as that seen in the original Dropout paper \cite{srivastava2014dropout}, with the exception of an $\tilde{A}$ instead on an $A$.
> >
> > Effectively the modifier matrix $H$ is re-scaling the input features scaling down entries with higher than average values and scaling up entries with lower than average values. While in a neural network dynamics are more complex and the activation outputs are evolving as training progresses we believe this adaptive structured dropout approach encourages the network to learn a more robust feature representation.
> >
> > We will these motivations and theoretical analysis to the Appendix in our paper.
> >
> > [1] A Simple Way to Prevent Neural Networks from Overfitting

---

### Author Response · Authors · 2022-11-15
**Motivation, Explanation and Theoretical Analysis**

We would like to provide two pieces of explanations from different angles to our work. This is also added to Appendix A of our paper.

### Approximating an Ensemble Using Dropout

From the initial Dropout paper we can consider unstructured Dropout as a way of training exponentially many sparse networks with extensive weight sharing and then at test time we approximately average this ensemble of networks.

ProbDropBlock is an adaptive and structured approach to Dropout. In ProbDropBlock instead of uniformly sampling sparse networks from the original network during training we sample sparse networks with lower activations more often.

If we assume activation saliency is correlated with the amount of information that is contained, the sampled sparse networks are now faced with harder problems since they are missing likely relevant information and so likely perform worse. And so by sampling these ‘worse’ performing sparse networks more often during training we provide additional training for these ‘worse’ sparse networks and improve the overall performance of the approximated average ensemble at test time.

Intuitively, we believe ProbDropBlock encourages the sampled sparse network to learn a more expressive feature map and not be overly dependents on particularly high output activations (features).

---

> ### Author Response · Authors · 2022-11-15
> **Motivation, Explanation and Theoretical Analysis (Cont.)**
>
> ### Marginalizing ProbDropBlock
>
> In considering the effect of our adaptive structured approach to Dropout we present a modified scenario of marginalising out Dropout in a simple linear regression case with a block size of 1. This setup was used in the original Dropout paper [1] to illustrate how Dropout acts as a regulariser when marginalised out.
>
> In our scenario we take as input the output activations $A \in \mathbb{R}^{N \times M}$ of a layer of a network, a weight vector $\mathbf{w} \in \mathbb{R}^{M}$ and a target vector of $\mathbf{y} \in \mathbb{R}^{N}$.
>
> As we all know the aim of linear regression is to minimise  $||\mathbf{y} - A\mathbf{w}||$. ProbDropBlock randomly drops entries in the matrix $A$ with entries being retained with a probability $p_{i,j} = 1 - (q \times \alpha_{i,j})$ where $\alpha_{i,j}$ gives the adaptive ratio for dropping feature entries based of relative magnitude. $p_{i,j}$ may be rewritten as $p \times h_{i,j}$ where $p$ is a constant keep probability and $h_{i,j}$ is a scalar that modifies p to give the keep probabilities for matrix entry $i, j$.
>
> After ProbDropBlock the input $A$ can be expressed as $R \ast A$ where $R \in \{0, 1\}^{N \times M}$ is a random matrix and $R_{i,j}$ $\sim$ Bernoulli$(p_{i,j})$ and $\ast$ represents element wise matrix multiplication. So marginalising the effect of dropout out of the objective function becomes
>
> \begin{equation}
>     \mathbb{E}_{R}
>     \left[
>     ||\mathbf{y} - \left( R \ast A\right) \mathbf{w}||^{2}
>     \right]
> \end{equation}
>
> This becomes
> \begin{equation}
>     \mathbf{y}^{T}\mathbf{y} -
>     \mathbf{w}^{T}\mathbb{E}\left[\left( R \ast A\right)^{T}\right]\mathbf{y} -
>     \mathbf{y}^{T}\mathbb{E}\left[\left( R \ast A\right)\right]\mathbf{w} +
>     \mathbf{w}^{T}\mathbb{E}\left[\left( R \ast A\right)^{T}\left( R \ast A\right)\right]\mathbf{w}
> \end{equation}
>
> It is clear that the $\mathbb{E}\left[\left( R \ast A\right)^{T}\right] = \mathbb{E}\left[\left( R \ast A\right)\right]^{T}$ and $\mathbb{E}\left[\left( R \ast A\right)\right] = P\ast A$ where $P$ is a matrix of the keep probabilities for each entry $(i,j)$. So $\mathbb{E}\left[\left( R \ast A\right)\right]$ may be rewritten as $p \tilde{A}$ where $\tilde{A} = H \ast A$, $p$ is a constant and $H$ is a re-scaling matrix of entries $h_{i,j}$ that modify the constant $p$ as needed.
>
> $\mathbb{E}\left[\left( R \ast A\right)^{T}\left( R \ast A\right)\right]$ is a little more involved but works out to be $p^{2}\tilde{A}^{T}\tilde{A} + (p - p^{2}) \text{diag}\left( \tilde{A}^{T}\tilde{A}\right)$.
>
> And so the full expectation works out to be
> \begin{equation}
>     \mathbf{y}^{T}\mathbf{y} -
>     p \mathbf{w}^{T} \tilde{A}^{T}\mathbf{y} -
>     p \mathbf{y}^{T} \tilde{A}\mathbf{w} +
>     p^{2} \mathbf{w}^{T}\tilde{A}^{T}\tilde{A}\mathbf{w} +
>     (p - p^{2}) \mathbf{w}^{T}\text{diag}\left( \tilde{A}^{T}\tilde{A}\right)\mathbf{w}
> \end{equation}
>
> Which can be rewritten as
> \begin{equation}
>     ||\mathbf{y} - p\tilde{A}\mathbf{w}||^{2} +
>     (p - p^{2})||\text{diag}\left( \tilde{A}^{T}\tilde{A}\right)^{\frac{1}{2}} \mathbf{w}||^{2}
> \end{equation}
>
> The second term is never negative as $p \ge p^{2}$ because $p \le 1$ and so the second term acts as a regulariser when we try to minimise this expression with respect to $\mathbf{w}$ as is the case in linear regression. In fact this equation takes the same form as that seen in the original Dropout paper \cite{srivastava2014dropout}, with the exception of an $\tilde{A}$ instead on an $A$.
>
> Effectively the modifier matrix $H$ is re-scaling the input features scaling down entries with higher than average values and scaling up entries with lower than average values. While in a neural network dynamics are more complex and the activation outputs are evolving as training progresses we believe this adaptive structured dropout approach encourages the network to learn a more robust feature representation.
>
> We will these motivations and theoretical analysis to the Appendix in our paper.
>
> [1] A Simple Way to Prevent Neural Networks from Overfitting

---

### Author Response · Authors · 2022-11-28
**Rebuttal acknowledgement and a quick summary of what we have done**

Dear reviewers,

Following up on our responses, we would like to quickly summarize that we have 1) provided a motivation and theoretical analysis of our proposed method and 2) presented more results and ablation studies as requested by some reviewers.

We would really appreaciate if we can have a fruitful discussion on this paper.

Thank you

---

### Author Response · Authors · 2022-12-09
**We are reaching the end of discussion period 2 (DDL December 12), and would really appreciate any futher feedback**

Dear reviewers,

As the deadline for discussion period 2 is approaching in 3 days, we would really appreciate any feedback on our responses to the questions and modifications of our paper.


Thanks.

---

### Decision · Program_Chairs · 2023-01-20

**Decision:**

Reject

**Justification For Why Not Higher Score:**

See the meta review and summary.

**Justification For Why Not Lower Score:**

N/A

**Metareview: Summary, Strengths And Weaknesses:**

In this paper, the authors propose a version of structured dropout called ProbDropBlock. They drop contiguous blocks from feature maps with a probability given by normalized feature salience values. All reviewers agree that revisiting structured dropout is a good research direction to continue for the deep learning community. However, the consensus among reviewers is that the improvements on MNLI and ResNet50 are too marginal and are not state-of-the-art results. Another main concern is that the motivation for assigning a higher drop rate to activations with a higher magnitude is unclear. Overall, the reviewers have seen merits in this paper, but because of the lack of empirical evidence, the panel's consensus is that this paper is not ready for ICLR 2023.

**Summary Of Ac-Reviewer Meeting:**

The consensus has been reached that due to the marginal improvements in the empirical results, the proposed approach is unlikely to create major impacts in the deep learning community.